Corrected: Author Correction

# Super-regional land-use change and effects on the grassland specialist flora

Alistair G. Auffret [1,2,3], Adam Kimberley [2], Jan Plue [2,4] & Emelie Waldén [2]

Habitat loss through land-use change is the most pressing threat to biodiversity worldwide. European semi-natural grasslands have suffered an ongoing decline since the early twentieth century, but we have limited knowledge of how grassland loss has affected biodiversity across large spatial scales. We quantify land-use change over 50–70 years across a 175,000 km$^2$ super-region in southern Sweden, identifying a widespread loss of open cover and a homogenisation of landscape structure, although these patterns vary considerably depending on the historical composition of the landscape. Analysing species inventories from 46,796 semi-natural grasslands, our results indicate that habitat loss and degradation have resulted in a decline in grassland specialist plant species. Local factors are the best predictors of specialist richness, but the historical landscape predicts present-day richness better than the contemporary landscape. This supports the widespread existence of time-lagged biodiversity responses, indicating that further species losses could occur in the future.

[1] Department of Ecology, Swedish University of Agricultural Sciences, Box 704475007 Uppsala, Sweden. [2] Biogeography and Geomatics, Department of Physical Geography, Stockholm University, 10691 Stockholm, Sweden. [3] Department of Biology, University of York, York YO10 5DD, UK. [4] School for Natural Sciences, Technology and Environmental Studies, Södertörn University, 141 89 Stockholm, Sweden. Correspondence and requests for materials should be addressed to A.G.A. (email: alistair.auffret@slu.se)

Habitat loss through land-use change is widely accepted as the primary driver of biodiversity decline worldwide[1,2]. Reductions in habitat availability are also limiting responses to climate change at both the population and community level[3,4]. In Europe, the biodiversity of multiple taxa is strongly dependent on semi-natural grasslands, characterised by a long history of traditional, low-intensity management[5,6]. However, agricultural intensification has resulted in many of these grasslands being converted to arable fields or degraded through grassland improvement or grazing abandonment, causing negative effects on biodiversity[7,8]. These processes still continue to the present day, reflecting the ongoing conversion of natural and semi-natural habitat occurring at the global scale[9,10].

Although habitat destruction is often associated with a direct loss of species, land-use patterns in the wider landscape can also contribute to determining biodiversity responses at the local scale. Varied landscapes can support larger species pools and additional populations of individual species, which can slow biodiversity loss in focal habitats[11]. As a result of processes occurring at landscape scales, communities can experience time lags of decades, or even centuries, before predicted future diversity losses[12,13]. Although the magnitude of both temporal delays and expected extirpations may depend on the types of habitat, landscape and organism being studied[8,13,14], these delays offer key opportunities to prevent further regional biodiversity losses through targeted restoration and conservation efforts[15,16]. However, widespread, directional land-use change results in landscape homogenisation (i.e. reductions in heterogeneity of habitat types within a landscape), weakening the landscape's mediating effects, with further detrimental implications for biodiversity and related ecosystem services[17–19].

Despite the build up of knowledge that has occurred through numerous studies of multiple landscapes and effects on biodiversity, there is still an urgent need to quantify land-use change and its effects on species across large spatial scales. In many cases, we still do not know the full extent of habitat loss, as a lack of historical land-use data means that quantifications of change are generally limited to the landscape scale, with analyses of land-use change at regional or larger scales being rare[8,20,21]. Coupling large-scale quantifications of land-use change with habitat-level species inventories is an important step forward in assessing the generality of land-use change and how local and landscape factors determine present-day biodiversity in landscapes spread across larger regions and with different historical contexts.

We quantify land-use change over 50–70 years in 6733 $5 \times 5$ km landscapes covering a 175,000 km$^2$ super-region covering southern Sweden. We identify changes in the cover of arable land, forest and open land (interpreted as being mostly semi-natural grassland, but also including wetlands and urban land uses, see Methods), alongside changes in landscape heterogeneity. We find that there have been significant losses in such open land, contributing to a widespread homogenisation of landscapes. Landscape information is then combined with plant species data from government-initiated inventories of 56 grassland specialist plant species in 46,796 semi-natural grasslands covering the study region. Splitting the grassland dataset in half, we use generalised linear models (GLMs) to identify how the past and present landscape, the local grassland conditions and regional differences explain the richness of grassland specialists on the set of training grasslands. Individual, single-predictor GLMs were first created to establish the predictive power of landscape variables including past and present land use and heterogeneity surrounding each grassland, before a final model was created including the most informative landscape variables along with the additional local variables describing the character of the focal grassland and its management. The relative and additive contributions of landscape, local and regional variables were assessed, before the power of the full model was then evaluated by predicting specialist species richness on the remaining validation grasslands. We find that local factors are the main drivers of grassland specialist richness, and there is a strong regional variation in both land-use change and effects on the specialist flora. Landscape variables are less important, but show that there is still an effect of historical land-use on present-day richness. We infer that the loss of open land, including much semi-natural grassland habitat, has already resulted in a large loss of grassland specialists, but that the role of the historical landscape implies that there could be possible future effects on biodiversity as a whole, and grassland specialists in particular, in the absence of conservation actions.

## Results

**Afforestation and widespread homogenisation.** Since the mid-twentieth century, the total cover of open land in southern Sweden has declined by 17%, with large regions showing strong declines at the landscape scale (Fig. 1a and Supplementary Table 1). This decline has contributed to a widespread landscape homogenisation, where more than two-thirds of the 6733 analysed landscapes have a lower landscape heterogeneity today than in the past (Fig. 1b). Land-use trajectories were strongly dependent on the characteristics of the landscape in the mid-twentieth century. Historically more open and forested landscapes both had relatively high forest cover in the historical maps, and both showed further increases in forest cover (Fig. 1c and Supplementary Fig. 1). Increased forest cover was especially pronounced in historically open areas, resulting in 94% of such landscapes becoming more homogeneous, compared to 72% of historically forested landscapes. In more agricultural areas, landscape heterogeneity was more likely to have increased, with the dominance of arable fields declining as some areas of arable land were converted into open land uses.

**Drivers of grassland specialist plant richness.** The 46,796 analysed grasslands contained a mean ± s.d. of 7.03 ± 4.27 of the 56 grassland specialists. The single-predictor GLMs showed that specialist species richness in the training dataset is positively affected by historical landscape heterogeneity and the degree of open land in the historical and contemporary landscape (Table 1). The cover of arable land surrounding the grassland was negatively associated with specialist richness for both the historical and contemporary landscape. Arable land cover in the past landscape was a better predictor of richness than arable cover today, whereas the reverse was true for open land use.

All landscape, spatial, local and management variables were found to have significant effects on grassland specialist richness (Table 2 and Supplementary Table 2). Local-level variables such as grassland area, heterogeneity and the presence of key grassland types had the strongest positive impacts on specialist richness in the full model, while the fraction of open habitat in the grassland and the present landscape had smaller positive effects. Grasslands with greater levels of the so-called improvement via fertilisation contained significantly fewer specialist species, as did those situated in landscapes with higher historical landscape heterogeneity, in contradiction with the single-predictor GLM. The number of specialist species in a grassland also depended quite strongly upon the region in which the grassland was situated. Landscape variables provided much lower predictive power than local variables and region both in terms of individual predictors in the full model and the explanatory power of the landscape model compared with the regional and the local model (Table 3 and Supplementary Tables 3–6). Nonetheless, all three groups of variables (landscape, local and region) provided significant

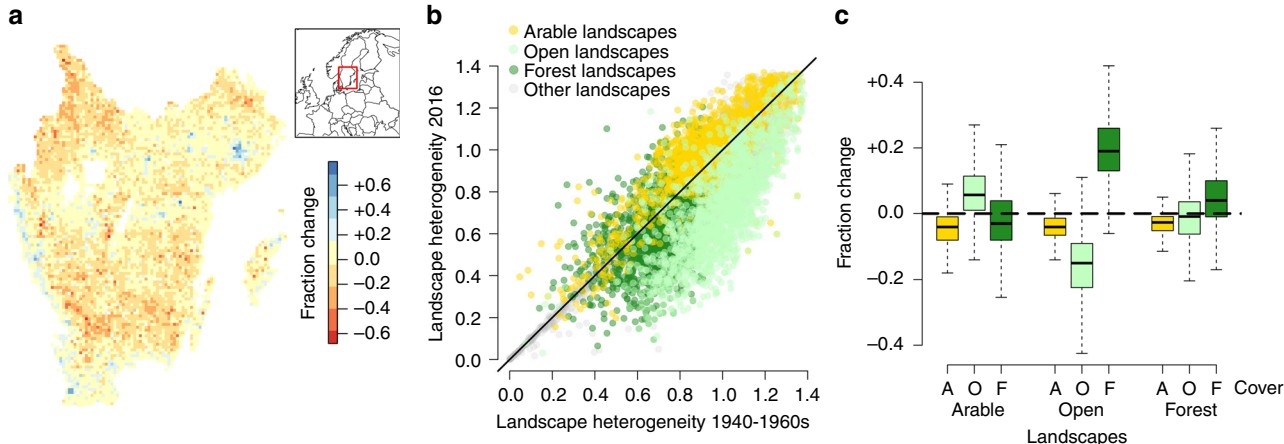

**Fig. 1** Landscape change in southern Sweden between 1940–1960s and 2016. **a** The extent of decline in open land in 6733 landscapes in the study region. **b** Comparison of historical and present-day landscape heterogeneity in all 6733 landscapes, with arable, open and forest landscapes defined as the top 25% 5 × 5 km grid squares with the highest cover of those categories in the historical maps (n = 1683). Black solid line at 1:1 between historical and present-day heterogeneity. **c** Change in cover of arable (yellow), open (light green) and forest (dark green) in arable, open and forest landscapes, showing median, interquartile range and range excluding outliers (defined as quartiles ± (1.5 × the interquartile range))

### Table 1 Effect of past and present landscape on grassland plant specialist richness

| Predictor | Coefficient | Lower | Upper | t | p | $R^2$ |
|---|---|---|---|---|---|---|
| Past landscape heterogeneity* | 0.065 | 0.026 | 0.104 | 3.251 | 0.0012 | 0.0004 |
| Present landscape heterogeneity | 0.022 | −0.011 | 0.055 | 1.310 | 0.1901 | 0.0000 |
| Past landscape open | 0.227 | 0.165 | 0.288 | 7.188 | <0.001 | 0.0022 |
| Present landscape open* | 0.472 | 0.414 | 0.530 | 15.930 | <0.001 | 0.0114 |
| Past landscape arable* | −0.138 | −0.178 | −0.098 | −6.732 | <0.001 | 0.0018 |
| Present landscape arable | −0.125 | −0.164 | −0.085 | −6.186 | <0.001 | 0.0015 |

Results of single-predictor generalised linear models on the effect of landscape variables on the number of grassland specialist plants in 23,398 Swedish semi-natural grasslands, including 95% confidence intervals. Asterisks indicate predictors carried forward to the full model, where inclusion of corresponding past and present data for landscape variables within the same model was not possible due to non-independence

additional power to explaining specialist richness (Table 3), although it was again local conditions that were found to make the most important contribution. The full model was able to correctly predict 34% of the variation in the specialist species richness of the validation half of the dataset (Fig. 2). The parameter estimate of 0.991 indicates a close relationship to 1:1, although Fig. 2 suggests a slight tendency toward over-prediction in the least species-rich grasslands.

## Discussion

Southern Sweden suffered an extensive loss of open land between 1940–1960s and 2016. Many landscapes lost open cover amounting to up to half of the total landscape (red pixels in Fig. 1a), reflecting the abandonment of semi-natural grasslands occurring across Europe during the twentieth century[8,22,23]. However, despite the strong losses of open land in historically open and forested landscapes, increases in some regions meant that the total reduction of open land was somewhat lower than estimates of semi-natural grassland loss from previous mapping studies[8,20]. Gains in open land were particularly prevalent in agricultural landscapes, which have seen abandonment in arable land uses across Europe[24]. The conversion of arable to open land is dominated by the creation of modern grassland on former arable fields, but also includes the spread of urban land uses[8], both habitat types being of considerably less biodiversity value than ancient semi-natural grasslands[25–27]. Increases in these habitat types will have offset other losses in open land, and we

therefore consider that the actual loss of semi-natural grassland habitat was likely much higher than the measured 17% reduction in open land (see e.g. Cousins et al. 2015[8]).

This loss of open, largely semi-natural grassland habitat appears to have had a substantial negative effect on the grassland specialist flora. The number of specialists present in remaining semi-natural grasslands is strongly coupled to grassland size and other characteristics such as degradation of habitat quality through grassland improvement. Coupled with the large observed decline in grassland habitat, particularly in historically open landscapes, this indicates that there have already been significant losses in grassland plant populations and possible local and regional species extirpations during the second half of the twentieth century. Moreover, the legacy of the historical landscape, predicting present-day diversity better than the modern landscape, indicate that further losses in grassland specialist species are likely to occur in the future without management intervention. Higher past landscape heterogeneity may have enabled the enduring persistence of larger landscape species pools that could temporarily support diversity in remaining grasslands[11,12]. Such lagged biodiversity effects are argued to present an opportunity for conservation interventions to prevent further declines[15,16], although much stronger local effects on specialist plant richness suggest that resources would be best spent on maintaining, improving and restoring existing and abandoned pastures and meadows (which would in many cases also increase landscape heterogeneity), rather than focussing on

**Table 2 Landscape, local and regional effects on grassland plant specialist richness**

| Variable | Para. Est. | Lower | Upper | t | p |
|---|---|---|---|---|---|
| Intercept | 2.028 | 1.988 | 2.067 | 100.526 | <0.001 |
| *Landscape* | | | | | |
| Present landscape open | 0.024 | 0.009 | 0.038 | 3.220 | 0.001 |
| Past landscape heterogeneity | −0.037 | −0.049 | −0.024 | −5.692 | <0.001 |
| Past landscape arable | −0.080 | −0.094 | −0.066 | −11.366 | <0.001 |
| *Local* | | | | | |
| Grassland area (log) | 0.178 | 0.162 | 0.195 | 20.863 | <0.001 |
| Grassland heterogeneity | 0.220 | 0.206 | 0.234 | 30.769 | <0.001 |
| Area Fennoscandian species-rich dry-mesic lowland grassland (log) | 0.254 | 0.241 | 0.267 | 38.445 | <0.001 |
| Area semi-natural dry grassland and shrubland on calcareous substrates (log) | 0.116 | 0.105 | 0.128 | 19.451 | <0.001 |
| Grassland open habitat | 0.069 | 0.055 | 0.082 | 9.730 | <0.001 |
| Grassland improvement | −0.111 | −0.125 | −0.096 | −14.958 | <0.001 |
| *Region* | | | | | |
| Parameter compared to baseline factor Kronoberg county, which had the greatest change (reduction) in open cover in the study region | Parameter estimate | | | | |
| | Mean = −0.174, Max = 0.109 (Uppsala), Min = −0.526 (Blekinge) | | | | |
| | *p* All <0.001, except Värmland, Kalmar, Västra Götaland and Gotland (n/s) | | | | |

Full generalised linear model explaining grassland specialist richness in 23,398 Swedish semi-natural grasslands including both landscape and local predictors along with region (county), including 95% confidence intervals. Full model output including values for all regions is available as Supplementary Table 2

**Table 3 Effect of landscape, local and regional variables on grassland plant specialist richness**

| | Adjusted $R^2$ | Residual deviance | DF | Deviance | p (>Chi) |
|---|---|---|---|---|---|
| *Effect of landscape* | | | | | |
| Landscape | 0.015 | | | | |
| Local + region | 0.316 | 39,072 | | | |
| Full model | 0.321 | 38,814 | 3 | 257.89 | <0.001 |
| *Effect of local* | | | | | |
| Local | 0.242 | | | | |
| Landscape + region | 0.104 | 50,789 | | | |
| Full model | 0.321 | 38,814 | 6 | 11,975 | <0.001 |
| *Effect of region* | | | | | |
| Region | 0.084 | | | | |
| Local + landscape | 0.245 | 42,924 | 14 | 4110.1 | <0.001 |
| Full model | 0.321 | 38,814 | | | |

Effect of groups of variables tested both through a comparison of adjusted $R^2$ of each model and $\chi^2$ tests comparing the full model (Table 2 and Supplementary Table 2) with models containing the other two groups of variables to assess if the addition of a group of variables results in a significant improvement of the model. Model descriptions of landscape, local, regional, landscape + regional, local + regional and landscape + local can be found in Supplementary Tables 3–6

improving the heterogeneity of the landscape per se. However, while plants are often found to be more strongly affected by local factors, landscape factors can be as or more important for richness in other more mobile organism groups such as invertebrates, mammals and birds[28].

Our results show strong regional differences in land-use change (Fig. 1) as well as variation in coefficient signs between the individual and full models (Tables 1 and 2). This provides further support to the stance that it is important to consider the impact of regional variation in large-scale studies of land use and land-use change[29], and to avoid careless extrapolation of measured biodiversity responses across multiple landscapes or regions. Agri-environmental schemes provide a good example where national or international guidelines for conservation management can sometimes be ineffective at local and landscape scales[30], and in some cases may even limit biodiversity for some organism groups[31,32]. Furthermore, our

results also indicate that historical context in the form of variation in both the pre-change landscape conditions and the trajectory of land-use change that has occurred can affect measures of biodiversity within a focal grassland, even in landscapes that may appear similar in the present day. Here, historical and present-day landscape factors provided relatively poor, yet significant explanatory power for grassland specialist richness compared to local factors. However, both these and other landscape-scale data, along with local factors such as grassland size and other site information, can be openly available from different sources, and can therefore prove extremely useful for understanding local patterns of biodiversity when site-by-site visits are not feasible[33,34]. This could then be useful for practitioners tasked with prioritising habitats and landscapes for conservation actions when applying national-scale policy recommendations into regional-level management practice.

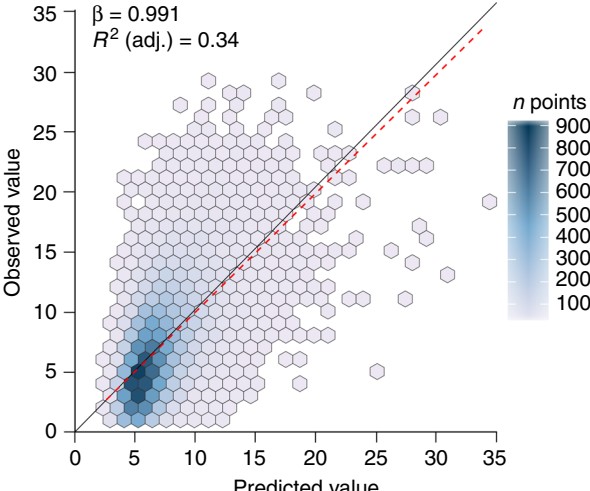

**Fig. 2** Performance of full model explaining grassland plant specialist richness. Predicted versus observed species richness of grassland specialists in the validation set of 21,018 Swedish semi-natural grasslands, using a full model containing landscape and local variables along with region. Points are binned into hexagons for clarity, given the large numbers of overlapping points in some areas. Black solid line at 1:1 between predicted and observed. Red dashed line is the regression line of observed values based on predicted

We present what we believe is—in terms of temporal and spatial extent and spatial resolution—the most extensive and rigorous quantification of twentieth century land-use change. Combined with a comprehensive database of semi-natural grassland inventories, we have identified the negative effects of widespread losses in grassland area and landscape heterogeneity on grassland specialist communities. It is clearly imperative that remaining semi-natural grassland habitat is protected and appropriately managed to promote biodiversity given alarming rates of continued habitat conversion[9,10], recognising regional variation in widespread landscape changes. However, it is also important to consider the landscape perspective, contributing both to biodiversity today, as well as buffering the loss of species following land-use change.

## Methods

**Data**. We used our own published digitisations of the Swedish Economic Map, which was created between the 1940s and 1960s over the 175,000 km² study area. Each map sheet covers 5 × 5 km and consists of a monochrome aerial orthophoto that was usually tinted green before arable fields, gardens and pasture that had recently been arable land were coloured yellow. Wooded areas and surface water show up as being darker on the underlying photograph, while lighter areas indicate open parts of the landscape, mainly consisting of grasslands, wetlands and urban land uses. Maps were digitised semi-automatically at a 1 m resolution, distinguishing between arable fields, forest and open areas, while surface water was then added from the open-access 2016 Swedish terrain map, resulting in four land-use categories. The historical period during which the mapping took place, along with the low population density of southern Sweden means that open areas can be broadly interpreted as managed grasslands, while the vast majority of land in the arable category was cropped arable land. Comparisons with manually digitised interpretations of the historical maps showed around very good agreement at the landscape level. For more information, see Auffret et al.[35,36]

To describe broad-scale land-use change over the whole study region, the 6733 digitised map sheets covering the study area were first aggregated to a 5 m resolution (package: raster; function: aggregate in the R statistical environment version 3.3.2 and above[37,38]), before proportions of the four land-use categories were calculated per sheet, and landscape heterogeneity calculated as Shannon diversity of these categories (package: vegan; function: diversity[39]). For present-day landscape information, the vector file of the 2016 Swedish terrain map was rasterised to the same resolution as the historical maps

(package: gdalUtils; function: gdal_rasterize[40]), before land-use categories were recoded into the broad categories of the historical map (see Supplementary Table 7) and semi-natural grassland habitat from the publicly available database of the national survey of Swedish semi-natural grassland habitats (TUVA—http://www.sjv.se/tuva) was added as open land. Proportion area and heterogeneity of the four land-use types were then calculated per map sheet as with the historical maps. To describe past and present land use in the landscape surrounding the focal grasslands, area of open land and heterogeneity were calculated within a 1000 m buffer surrounding each grassland for both the historical landscape and the present day (buffer drawn using package: rgeos; function: gBuffer[41]).

We used species and habitat data from the Swedish semi-natural grassland database. The survey is intended to be a continuous monitoring project, although the majority of the grassland patches have only been inventoried once, during the project's initial period 2002–2004[42]. The whole of each inventoried grassland was surveyed for the presence and relative abundance of 56 grassland specialist species, which are vascular plants typical for Swedish semi-natural grasslands (see Supplementary Table 8). In addition to species occurrences, inventories involved a statement as to whether or not the grassland is currently managed, as well as visual estimations of the fraction cover of the grassland that has been subject to grassland improvement (including fertilisation, seed sowing and other ground preparation), the fraction of open habitat within the grassland (i.e. not covered by trees or shrubs) and the fraction cover of 30 categories of habitat from the EU Habitat Directive (92/43/EEC; plus two additional categories for 'other' and 'mosaic' landscapes, examples include Fennoscandian wooded meadows and Boreal Baltic coastal meadows, see Supplementary Methods). These grassland sub-habitats were converted into a measure of grassland heterogeneity using Shannon diversity, as above. Finally, the geographic information system (GIS)-derived total area (ha) of the grassland is also included in the database. We included all 46,796 grasslands that are classed as being currently managed, and in which at least one of the 56 grassland specialists was observed.

**Analysis**. Changes in fractional cover of arable, open and forest land use were calculated per map sheet by subtracting the values of the historical data from the values of the contemporary data. To assess change in different types of landscape, map sheets with the top 25% cover (i.e. 1683 sheets) of arable, open and forest in the historical dataset were assigned as being characteristic of those land-use types. Total percentage change of open land across the whole region was also calculated.

The analysis of plant specialist richness in managed semi-natural grasslands consisted of three steps. For all steps, we used the same random selection of 23,398 grasslands from the full dataset (R-function: sample). These training grasslands were used to build all statistical models. The remaining validation grasslands ($n$ = 23,398) were reserved and used to assess the ability of the final model to predict the number of grassland specialist species within valuable grasslands across Sweden. Training and validation grasslands were similar in terms of number of specialist plant species (mean ± s.d. training grasslands: 7.01 ± 4.21; validation grasslands: 7.05 ± 4.23).

Our first step was to create six GLMs (R-function: glm) to establish the predictive power of each individual landscape variable in predicting specialist species richness in the training grasslands. In each model, species richness was the response variable, with one of past landscape heterogeneity, present landscape heterogeneity, past landscape open cover, present landscape open cover, past landscape arable cover and present landscape arable cover as the predictor variable. Past and present landscape forest cover were not used as predictors in models due to the strong negative correlation between open and forest cover (Supplementary Table 9), and our focus on grasslands mean that we analysed open cover only. These and all subsequent GLMs were fitted with Quasi-Poisson distributions to correct for overdispersion in the response variable. The adjusted $R^2$ for each model was calculated using the function rsq from the package rsq[43].

Our next step was to evaluate the relative importance of landscape variables, local variables describing the focal grassland and its management and region in explaining grassland specialist species richness. To do this, we fitted seven GLMs. The first was a landscape GLM containing predictor variables from the previous step. Due to non-independence of corresponding past and present landscape variables (Supplementary Table 9), we chose to include the variables with the highest predictive power based on the single-predictor GLMs (Table 1). The landscape model therefore contained past landscape heterogeneity, past landscape arable cover and present-day landscape open cover as predictor variables. A second, local GLM was then created, including the following grassland-level variables: grassland area (log-transformed to reduce the skew in its distribution), grassland heterogeneity, fraction improved grassland, fraction open habitat, logged area of semi-natural dry grassland and shrubland on calcareous substrates and logged area of Fennoscandian species-rich dry-mesic lowland grassland. The two grassland sub-habitats had been determined by an exploratory random forest analysis[44] to be the most important of the Habitat Directive habitats in predicting grassland specialist species richness (Supplementary Methods). A third, region model had only the administrative region (county) of Sweden in which the grassland was located as a predictor variable. Region is likely to represent various aspects of policy history, regional

geography and species pools that are likely to differ across large spatial scales. Additionally, the historical maps were created on a county-by-county basis and differ in the time that land use was mapped. The fourth, fifth and sixth models contained two of each of the sets of variables, that is, landscape + local, landscape + region and local + region. The seventh model was a full model containing all landscape, local and region variables. To facilitate comparisons between parameter estimates, predictors in all seven models were standardised to have a mean of 0 and a standard deviation of 0.5 (package: arm; function: standardize[45]). To compare the relative effect of landscape, local variables and region in predicting specialist plant species richness, adjusted $R^2$ of each model was calculated. Then, $\chi^2$ tests were performed to evaluate the additional explanatory power that each set of variables contributed, with each of the landscape + local, landscape + region and local + region compared to the full model to test the contribution of region, local and landscape values, respectively (R-function: anova).

Our third and final step was to use the full model to predict the number of grassland specialists in the 23,398 validation grasslands that were withheld from the modelling. The predicted values were used as a predictor in a linear regression (R-function: lm) with observed numbers of grassland specialists as the response, as recommended by Piñeiro et al.[46].

**Data availability**. Historical land-use data are available at https://doi.org/10.17045/sthlmuni.4649854. Present-day open-access terrain maps are available from the Swedish Agency Lantmäteriet (https://www.lantmateriet.se/sv/Kartor-och-geografisk-information/oppna-data/). GIS data for the grassland database can be downloaded from the Swedish Agency Jordbruksverket (http://www.sjv.se/tuva), where species occurrence data per grassland can also be browsed online. The full database is available as a spreadsheet on request from the agency.

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

## Acknowledgements

This work would not have been possible without the efforts of the practitioners who inventoried tens of thousands of semi-natural grasslands across Sweden. We also thank Simon Jakobsson, Helle Skånes, Marika Wennbom and Heather Wood for their help in digitising the historical maps, and to the maintainers of University of York's Advanced Research Computer

Cluster (YARCC) which was used for the quantification of land-use change. A.G.A. and E.W. are supported by the Swedish Research Council Formas projects 2015-1065 and 2009-1105, while A.K. is supported by The Swedish Research Council Vetenskapsrådet project E0526301. J.P. is funded by The Foundation for Baltic and East European Studies.

## Author contributions

All authors contributed equally to this study. J.P. and A.G.A. conceived the broad study idea, with all authors working together to design research questions, plan analyses and interpret results. Data were collated by E.W. and A.G.A. and analysed by A.K. and A.G.A. A.G.A. led the writing, with significant input from all co-authors.

## Additional information

**Competing interests:** The authors declare no competing interests.

