## [Peer Review File · Nature Communications]

Reviewers' comments:

Reviewer #1 (Remarks to the Author):

Review

Past habitat loss and land-use change threaten plant biodiversity

Auffret et al.

Auffret et al. have carried out an impressive study with considerable spatial (175 000 km²) and temporal coverage (land-use change over 50-70 years), as well as with massive biodiversity data (current specialist plant data from 46 796 grasslands). They detected substantial changes in land-use patterns in Sweden, and massive homogenisation of historically 'open' landscapes, while arable landscapes had turned more heterogeneous over past 50-70 years. In addition, they disentangled factors determining current biodiversity of grasslands and detected presence of time-lags.

I consider the contribution an important addition to current knowledge, especially due to its extensive temporal and spatial scale. Paper is well-written, interesting to read and provides new information regarding the land-use changes over past decades in Northern Europe although main findings largely confirm the patterns that have been previously detected in smaller scales. Statistical analysis are sound and appropriate.

I also believe there is room for additional improvement to strengthen the conclusions. For example, the specifics of data that the manuscript is based on can be explained in more detail. Supplementary information could provide examples of 'typical' historical vs. present landscapes. More information about what is exactly covered in historical and present data under different land-use types would be highly beneficial. For example, what constitutes as 'arable' in historical data? Does it cover only ploughed areas and/or permanent crops or are there also fallow lands or perhaps even some grazed areas? What are relative and absolute values of different land-cover types? It would be also very informative if more details would be provided regarding the changes in land-use. What are absolute values of change and what have been the main drivers of land-use change? It is visible from Figure 1c that the land-use change has been especially pronounced for historically forested areas. What is the background of these patterns?

Some methodological challenges (that always accompany this kind of ambitious data) can also be further mitigated. More specifically, land-use category 'open' consists of variety of habitats for both historic and present land-use. Authors point out that although there are also wetlands and urban areas included, the land-use category 'open' consist mainly of grasslands. However, due to the lack of further explanation about the specifics of data, it remains unresolved to what extent the dynamics of broad category 'open' allow to draw assumptions about the fate of grasslands in the region. Currently, authors found 17% decline in total cover of open land in southern Sweden "predominantly driven by a loss of semi-natural grassland cover". Due to very loose definition of 'open' areas both historically and currently (involving urban areas, parks etc), linking dynamics of 'open' areas with loss of semi-natural grasslands can potentially lead to strong underestimation of actual grassland losses, as any land conversion from semi-natural grasslands to other open areas is not taken into account. It would be useful – if deemed necessary – to provide some estimate of 'real' cover change in semi-natural grassland area e.g. by quantifying it out of all open areas in a subset of randomly selected cells.

As a minor note, areas of 'semi-natural dry grassland and shrubland on calcareous substrates' and of 'Fennoscandian species-rich dry-mesic lowland grassland' from TUVa database are considered in models focusing on 'Local and management' variables. Why they are considered as local factors while other landscape variables (including grassland area) are considered under variable group 'Landscape and spatial'? Would it make a difference in the results if they are also considered together with other landscape variables?

In conclusion, I believe the study provides a lot of valuable information regarding magnitude of land -

use change in Europe and its impacts on grassland biodiversity. Important message about the effects of past land-use on grassland biodiversity has considerable conservation implications.

Reviewer #2 (Remarks to the Author):

March 28, 2018.

Overall comments:

In this interesting study, the authors digitize land cover over a huge area of southern Sweden in the present and 40-70 years in the past. This type of historical land cover data for such a wide extent is rarely available – even if the aerial photos or maps with which to quantify it are available, the enormous task of digitizing them is usually not taken on. In addition, the study was based on maps with an astoundingly high resolution (1 metre). The results show that overall there has been a 17% loss of 'open' land cover, but that the degree of loss varies depending on the starting conditions of the landscape, with the greatest losses of open land having occurred in landscapes that began as mostly forested. In addition, some areas have actually seen an increase in open land cover types.

The authors then test for correlations between surrounding landscape composition and heterogeneity in the past and present and the local species richness of grassland specialist plants using an enormous dataset of over 40,000 surveyed semi-natural grasslands. I know of a several studies that have quantified such correlations between landscape context and local plant communities, but none with this many data points or at this large spatial extent! The results suggest that, while local factors are the strongest predictors of grassland specialist species richness, the percent of arable land within 1000m of a grassland in the past, the heterogeneity of land cover within 1000m in the past, and the current amount of open cover within 1000m are also correlated with this response variable. No study that I know of has been able to test the role of the surrounding landscape on plant community composition at such a massive scale, with such a huge dataset, and with such high-quality historical land cover data, and as such I think this is an interesting and useful study.

I have two larger concerns about the work, as follows:

(1) Regarding the statistical approach:

(a) the authors include in a set of variables they call "landscape and spatial" predictors two factors which to me are clearly LOCAL: the area of the grassland patch surveyed, and the heterogeneity of grassland types within that patch.

(b) the authors show that a large model including both sets of predictors (what they call "landscape and spatial" and what they call "local and management" predictors) explains more variation in the richness of grassland specialists than either set alone. The full model explains an additional 8.5% of the residual deviance of either set alone. However, the analysis provides no way for the reader to gauge how much the inclusion of the landscape variables (which to me should include ONLY landscape composition and heterogeneity within the 1000m buffer) really improves the model after the local variables and the categorical region factor are accounted for. I would also like to know how much variation 'region' explains on its own. There are a few ways to do this. What I would like to see is either or both of the following options: (1) compare the AIC or r-squared for a model with region only, a model with local variables only (grassland area, within-grassland heterogeneity, area of the 2 specific fine-scale grassland types, grassland openness, and grassland improvement), a model with local variables + region, and the full model with local variables + region + landscape variables. (2) perform the regression shown in Figure 2 using the withheld test data for each of these different models. It seems that, while significant, the influence of the surrounding landscape cover and

heterogeneity is rather low once local factors are accounted for. This is important when considering conservation implications, because for example, if the data show that the local factors and region alone predict the species richness of grassland specialists with an R-squared of 0.32, and adding the landscape factors improve this only very slightly, say to 0.34, management at the local scale will be more effective for conserving these species.

(c) The authors do not specify if and how they tested for pairwise correlations between all the predictors in the 'full' model. I'm wondering if high collinearity between past landscape heterogeneity and one of the other predictors is the reason why the sign of the coefficient for this factor switches when it is included in the big model with all the other factors. The authors note this switch but do not offer an explanation for it. Might there also be interactions between 'region' and one or more of the other predictors? The authors didn't say if they considered potential interactions.

(2) Regarding use of the term 'biodiversity'

"Biodiversity" is a ubiquitous term in ecological research but it is usually poorly defined. Throughout the paper the authors make conclusions and interpretations about 'biodiversity' when what they are actually measuring is 'species richness of grassland specialist plants'. This may seem trivial, but it is actually very important. If we really want to understand how landscape change is affecting plant and animal communities, we need to be specific about what we are measuring. In this case, we do not know how the changes on the landscape have related to 'plant biodiversity', because that would include ALL plant species, and would involve measuring not only taxonomic, but functional and genetic diversity as well. By choosing only grassland specialist species, the authors are making a value judgement up front that these are the particular species we should care about most. This should be made clear throughout the manuscript, preferably by avoiding the term biodiversity in favour of what was actually measured: richness of grassland specialist plants.

Line by line comments:

Title: the title is not specific enough, and also misleading. Your data do not show that past habitat loss threatens plant biodiversity. Your data show that the species richness of grassland specialist plants is positively correlated with past landscape heterogeneity. I object to using the term 'biodiversity' when what the study has tested is species richness of semi-natural grassland specialist vascular plants. See overall comments above. The title should emphasize the true novelty of this study, which is assessing past and present landscape composition and heterogeneity at an unprecedented spatial scale, and testing for relationships between landscape composition/heterogeneity and plant community data from a massive set of on-the-ground surveys. Suggested titles:

Land-use change across a super-region and links to the diversity of grassland specialist plants
Super-regional land-use change over several decades and its effects on the grassland specialist flora
Quantifying land-use change across a super-region and testing its effects on the grassland flora

Abstract:

Line 38: a decline in grassland specialist plant species. There might actually have been an INCREASE in total plant species richness (due to colonization by disturbance tolerant species, or weeds associated with crops or urban land use). It is important to be clear about this.

Line 39: again 'biodiversity' is misleading. "present-day richness of grassland specialists" is honest.

Main:

Line 72-76: Yes – this is well put and aptly describes the need for quantifying historical land-use/land-cover change at regional spatial extents.

Line 87: Why "landscape and spatial"? Why not just "landscape"? The term 'spatial' to me implies some kind of spatial autocorrelation analysis, which is not included here.

Line 91-94: again, 'biodiversity' is misleading. Suggest "diversity of grassland specialists".

Line 96-98: it is a little strange to me that the 'open' category includes both grasslands AND urban land use – which seem quite starkly contrasted to me – but not arable land (which also, in many people's understanding, is 'open'). It might help to more explicitly define 'open land' where it is introduced, around line 79. Here, it would be easier for me to understand what is going on if you explained what replaced the open land. e.g. is it "... predominantly driven by a loss of semi-natural grassland cover due to conversion to arable land and afforestation" ?

Line 103: "This is due to a decline in arable land at the expense of more modern grassland and urban land uses". How can you tell this from your data, which lumps modern grassland, semi-natural grassland, and urban land use all together in the 'open' category?

Figure 2: Is the beta reported correct? The number given in the text is different.

Line 130: "by a very similar amount". This is a bit vague. It looks to me like including all the variables explains a further 8.5% of the deviance explained by either local or landscape sets alone. I really think Supplementary Table 4 should be presented in the main text.

Line 134: parameter estimate here in text given as: 0.987. In Figure 2 it is given as: 0.991.

Line 138: "The loss of semi-natural grassland habitat has had a negative effect on biodiversity". This is not accurate or precise. Your data show: OPEN habitats (including semi-natural grasslands, but also, wetlands, urban land use) have declined. Your data show: semi-natural grassland patches surrounded by more 'open' land use tend to have higher numbers of grassland specialist species (although, the amount of variation explained is small in comparison to the effect of the AREA of the semi-natural grassland patch AND you don't actually have a measure of change in the number of grassland specialist species over time). Your statement should be: "The loss of open habitats may lead to a decline in the species richness of grassland specialist plant species in semi-natural grasslands."

Line 144: present day richness of grassland specialists (NOT biodiversity). Also – yes I agree that the past landscape heterogeneity was a stronger predictor of grassland specialist species richness, but BOTH were quite weak compared to the effect of grassland area and grassland heterogeneity.

Therefore, I expect it would be better (not to mention easier!) to manage for larger, more heterogeneous grassland patches, rather than trying to manage heterogeneity on a landscape scale!

Line 152: Yes, this change in sign of the correlation between landscape heterogeneity and grassland specialist species richness when in the full model compared to when alone is interesting. Is it because past landscape heterogeneity is correlated with one of the other predictors? (so the sign of the coefficient changes when the correlated predictor is included?). Or is it because the effect of heterogeneity is different between regions, and perhaps an interaction term should have been included? I would like to know how much correlation there is between all the pairs of predictor variables.

Line 156-163: I find this whole section very vague. HOW could this lead to "inaccurate or erroneous policy decisions and misdirected conservation management efforts"? Can you provide an example?

Line 159: "Local factors were shown to be most important predictors of grassland diversity, but we found that landscape and spatial data can provide useful local context and are more straightforward to collect at large spatial scales when site-by-site visits are not feasible". Okay, but two of your "landscape" factors (in fact, the 2 with the strongest influence) are in my opinion actually "local" factors – the area of the grassland patch itself, and habitat heterogeneity within that patch. See my comment above re: presenting results in a way that allows us to measure the importance of the landscape-scale openness, heterogeneity, or arable cover once these local variables have been accounted for.

Line 161: "Such information could be a useful first step in consolidating national-scale policy with appropriate local-scale conservation practice." Again, it is very unclear to me what you are suggesting here. Earlier in this paragraph you stated that places with different historical landscape conditions and landscapes in different REGIONS changed in different ways – which makes it dangerous to extrapolate patterns across different historical conditions or different regions. Now you are saying that national-scale policy should be consolidated. Do you mean that national-scale policy needs to be tailored such

that different conservation measures are promoted in different regions and/or for landscapes with different historical conditions? On the other hand, in the previous paragraph you suggest that higher landscape heterogeneity ACROSS THE WHOLE REGION could promote richness of grassland specialists. So, should national-scale policy aim to increase landscape heterogeneity everywhere? Line 169: "It is clearly imperative that remaining semi-natural grassland habitat is protected and appropriately managed to promote biodiversity given alarming rates of continued habitat conversion." Okay, it is clearly imperative that IF we want to maintain richness of grassland specialists in semi-natural grasslands, THEN we should protect and manage these remaining grasslands. But I don't think your scientific results lead to this conclusion – this is a value judgement.

Tables:

Table 1: the asterisk should be beside "Present landscape open" instead of "Past landscape open". Again, seems to me the area of the grassland patch, and the heterogeneity within this patch are actually 'local' predictors...

Table 2: change "species richness" in caption to "species richness of grassland specialists". Also, the caption should note that region (county) is also included in the model.

Methods:

Line 325: was each grassland surveyed exhaustively, or did they just do surveys of PLOTS within the grasslands? e.g. this is total richness of grassland specialists in the whole grassland patch, or 'species density' of grassland specialists within a consistent same-sized survey plot? I think it is the former but it would be good to specify.

Line 330: how could surveyors know what portion of the grassland had been fertilized or had seeds added? Did they interview land managers?

Line 331: "the fraction of openness in the grassland" – this is very confusing to me, especially when the term 'open land' is already being used on a regional scale to indicate grassland/wetland/urban land. By "fraction of openness" do you mean the fraction that is not invaded by woody trees or shrubs? It would be good to clarify this.

Line 331: "30 categories of habitat"... so are these essentially finer-scale habitat definitions within the 'open' category? It might help to give a few examples (and reference Supplementary Table A1.

Line 335: by "indicator species", do you mean, at least one of the 56 semi-natural grassland specialists?

Line 345: "plant biodiversity" – see overall comments. This study is not analyzing plant biodiversity. Could change to "diversity of grassland specialists", or "species richness of semi-natural grassland specialists"

Line 352: shouldn't the number be 8, not 6, for the single-predictor models? Predictors are: (1) area of the surveyed grassland, (2) habitat heterogeneity of the surveyed grassland, (3) present landscape heterogeneity within 1000m buffer, (4) past landscape heterogeneity within 1000 m buffer, (5) present percentage open cover within 1000m buffer, (6) past percentage open cover within 1000m buffer, (7) present percentage arable cover within 1000m buffer, (8) past percentage arable cover within 1000m buffer

Line 354: specialist species richness

Line 359: which type of r-squared did you use, and was it adjusted r-squared? Did you consider presenting the AIC for each model? What is the R2 (or AIC) of a null model with intercept and region only?

Line 369: it is confusing here how 'landscape' versus 'local and management' were defined. If I understand correctly, the total area of the surveyed grassland, and the heterogeneity of different fine-scale habitat types within that patch, were considered 'landscape'. But then the area of two specific fine-scale habitat types within that patch ((1) semi-natural dry grassland and shrubland on calcareous substrates and (2) Fennoscandian species-rich dry-mesic lowland grassland) were considered 'local

and management' predictors.

Why would not grassland area and heterogeneity within the grassland be considered 'local'?

Also – wouldn't there be a negative correlation between the area of the 2 fine-scale grassland types (e.g. Fennoscandian species-rich dry-mesic lowland grassland) and fraction improved grassland? If there is a greater area of improved grassland, wouldn't there be less area of these specific semi-natural grassland types? Did you check for collinearity between the predictors before putting them all into the one large model?

Line 382: "Chi-squared tests were then performed to evaluate the additional explanatory power contained within the third, full model compared to the first landscape and spatial and the second local and management models (R function: anova)." I don't see these results reported anywhere... oh, now I see it's in Supplementary Table 4. I think this table should be in the main text.

Supplementary Information:

Line 19: insert "and" before "total grassland area"

Table A1: Alkaline fens

Supplementary Table 4: I really think this should be presented in the main text.

Supplementary Table 6:

Epipactis palustris

Hypochaeris maculata

Response to reviewers

Reviewer #1 (Remarks to the Author):

Auffret et al. have carried out an impressive study with considerable spatial (175 000 km²) and temporal coverage (land-use change over 50-70 years), as well as with massive biodiversity data (current specialist plant data from 46 796 grasslands). They detected substantial changes in land-use patterns in Sweden, and massive homogenisation of historically 'open' landscapes, while arable landscapes had turned more heterogeneous over past 50-70 years. In addition, they disentangled factors determining current biodiversity of grasslands and detected presence of time-lags.

I consider the contribution an important addition to current knowledge, especially due to its extensive temporal and spatial scale. Paper is well-written, interesting to read and provides new information regarding the land-use changes over past decades in Northern Europe although main findings largely confirm the patterns that have been previously detected in smaller scales. Statistical analysis are sound and appropriate.

Many thanks for the positive evaluation of our manuscript!

I also believe there is room for additional improvement to strengthen the conclusions. For example, the specifics of data that the manuscript is based on can be explained in more detail. Supplementary information could provide examples of 'typical' historical vs. present landscapes. More information about what is exactly covered in historical and present data under different land-use types would be highly beneficial. For example, what constitutes as 'arable' in historical data? Does it cover only ploughed areas and/or permanent crops or are there also fallow lands or perhaps even some grazed areas? What are relative and absolute values of different land-cover types? It would be also very informative if more details would be provided regarding the changes in land-use. What are absolute values of change and what have been the main drivers of land-use change? It is visible from Figure 1c that the land-use change has been especially pronounced for historically forested areas.

What is the background of these patterns?

We have now added more detail to address these concerns.

- In the methods section, we now provide much more information on the historical maps that we used, the digitisation method used to produce the historical land-use data and that the data were validated against detailed independent digitisation. We describe that the map was based on a aerial

orthophoto onto which arable land use was added, with open and forested areas being visible on the underlying photograph. We describe that arable land also includes gardens, and some grazed areas on former arable fields (**Lines 237-252**). However, because of the timing of the maps being created, and the low population density of Sweden, we state that the vast majority of the arable land-use class is in fact arable fields. Similarly, we do not believe that urban land uses will have had more than a negligible contribution to open in land-use class in any but the handful of maps showing the central areas of Sweden's main cities.

- We have expanded the results to give some more details about change in cover and heterogeneity (**Lines 116-122**). There is a new table detailing the change in cover and heterogeneity of the whole region and the different landscape categories (**Supplementary Table 1**). We have also added more details regarding the nature and drivers of land-use change in the discussion (**Lines 173-185**).

- We have added a new supplementary figure (**Supplementary Figure 1**) that uses the layout of Figure 1c (change in different types of landscapes) to show what the subgroups of Arable landscapes, Open landscapes and Forested landscapes looked like in terms of land use both in the past and today. We also show example maps from each of these three categories, which gives the reader some extra information about what the input data looked like.

- We have not added any information regarding the absolute levels of change in land use across the study region, but rather stick with using the absolute level of (fraction) change within landscapes as the basis of Figure 1a and 1c, and relative fraction change for summary data and the new **Supplementary Table 1** (e.g. 17% reduction in open cover). We do not think it would be helpful to report statistics about N-thousand square kilometres of open land lost, because we know that there have been losses and gains in all land-use categories, and so we know that these figures would not be accurate.

Some methodological challenges (that always accompany this kind of ambitious data) can also be further mitigated. More specifically, land-use category 'open' consists of variety of habitats for both historic and present land-use. Authors point out that although there are also wetlands and urban areas included, the land-use category 'open' consist mainly of grasslands. However, due to the lack of further explanation about the specifics of data, it remains unresolved to what extent the dynamics of broad category 'open' allow to draw assumptions about the fate of grasslands in the region.

Currently, authors found 17% decline in total cover of open land in southern Sweden “predominantly driven by a loss of semi-natural grassland cover”. Due to very loose definition of ‘open’ areas both historically and currently (involving urban areas, parks etc), linking dynamics of ‘open’ areas with loss of semi-natural grasslands can potentially lead to strong underestimation of actual grassland losses, as any land conversion from semi-natural grasslands to other open areas is not taken into account. It would be useful – if deemed necessary – to provide some estimate of ‘real’ cover change in semi-natural grassland area e.g. by quantifying it out of all open areas in a subset of randomly selected cells.

We think that our additions to the manuscript based on the comments above are also relevant here, namely more details of the historical maps and how they were interpreted, and in our discussion of the results. Although we agree that it would be nice to know the extent of semi-natural grassland loss from our maps, we do not think there is an acceptable way to do this. The idea of this manuscript is to look at broad-scale change over a very large area. Quantification of a subset of pixels or maps would take our large-scale analysis down to a much smaller scale, and the large differences in change across regional and historical landscape types shown in Figure 1 means that such a quantification could not possibly be scaled up to reflect the region as a whole. We agree that the real loss of semi-natural grassland cover was likely much higher than 17%, and have discussed this more thoroughly in the context of agricultural change in Europe in the text (**Lines 173-185**).

As a minor note, areas of 'semi-natural dry grassland and shrubland on calcareous substrates' and of 'Fennoscandian species-rich dry-mesic lowland grassland' from TUVa database are considered in models focusing on ‘Local and management’ variables. Why they are considered as local factors while other landscape variables (including grassland area) are considered under variable group ‘Landscape and spatial’? Would it make a difference in the results if they are also considered together with other landscape variables?

Our initial idea with the division of variables into the two categories ‘landscape and spatial’ and ‘local and management’ was that ‘landscape and spatial’ variables were those that could be created from a computer, i.e. by using digital maps to measure the size of a grassland and gauge the local and landscape heterogeneity from existing sources. ‘Local and management’ variables were those that required more specific data about each grassland and would require a site visit. We wanted to see to what extent the two groups could separately or together predict the richness of grassland specialist plants, which would be useful information for researchers and practitioners. We admit that

this was not particularly well explained, and drew questions from both referees. We have therefore split the variables into local (size of grassland and everything within it) and landscape (everything in the 1km landscape surrounding the grassland). This is more intuitive both linguistically and with regard to the main focus of the study, i.e. land-use change and effects on biodiversity (see e.g. **Table 2**).

In conclusion, I believe the study provides a lot of valuable information regarding magnitude of land-use change in Europe and its impacts on grassland biodiversity. Important message about the effects of past land-use on grassland biodiversity has considerable conservation implications.

Thank you

Reviewer #2 (Remarks to the Author):

Overall comments:

In this interesting study, the authors digitize land cover over a huge area of southern Sweden in the present and 40-70 years in the past. This type of historical land cover data for such a wide extent is rarely available – even if the aerial photos or maps with which to quantify it are available, the enormous task of digitizing them is usually not taken on. In addition, the study was based on maps with an astoundingly high resolution (1 metre). The results show that overall there has been a 17% loss of ‘open’ land cover, but that the degree of loss varies depending on the starting conditions of the landscape, with the greatest losses of open land having occurred in landscapes that began as mostly forested. In addition, some areas have actually seen an increase in open land cover types.

The authors then test for correlations between surrounding landscape composition and heterogeneity in the past and present and the local species richness of grassland specialist plants using an enormous dataset of over 40,000 surveyed semi-natural grasslands. I know of a several studies that have quantified such correlations between landscape context and local plant communities, but none with this many data points or at this large spatial extent! The results suggest that, while local factors are the strongest predictors of grassland specialist species richness, the percent of arable land within 1000m of a grassland in the past, the heterogeneity of land cover

within 1000m in the past, and the current amount of open cover within 1000m are also correlated with this response variable. No study that I know of has been able to test the role of the surrounding landscape on plant community composition at such a massive scale, with such a huge dataset, and with such high-quality historical land cover data, and as such I think this is an interesting and useful study.

Thank you

I have two larger concerns about the work, as follows:

(1) Regarding the statistical approach:

(a) the authors include in a set of variables they call “landscape and spatial” predictors two factors which to me are clearly LOCAL: the area of the grassland patch surveyed, and the heterogeneity of grassland types within that patch.

Please see our response to Referee 1, above. We have changed our groupings now to test local vs. landscape drivers of specialist richness, rather than GIS (landscape and spatial) vs. site visit (local and management). See e.g. **Table 1 & 2**.

(b) the authors show that a large model including both sets of predictors (what they call “landscape and spatial” and what they call “local and management” predictors) explains more variation in the richness of grassland specialists than either set alone. The full model explains an additional 8.5% of the residual deviance of either set alone. However, the analysis provides no way for the reader to gauge how much the inclusion of the landscape variables (which to me should include ONLY landscape composition and heterogeneity within the 1000m buffer) really improves the model after the local variables and the categorical region factor are accounted for. I would also like to know how much variation ‘region’ explains on its own. There are a few ways to do this. What I would like to see is either or both of the following options: (1) compare the AIC or r-squared for a model with region only, a model with local variables only (grassland area, within-grassland heterogeneity, area of the 2 specific fine-scale grassland types, grassland openness, and grassland improvement), a model with local variables + region, and the full model with local variables + region + landscape variables. (2) perform the regression shown in Figure 2 using the withheld test data for each of these different models. It seems that, while significant, the influence of the surrounding landscape cover and heterogeneity is rather low once local factors are accounted for. This is important when

considering conservation implications, because for example, if the data show that the local factors and region alone predict the species richness of grassland specialists with an R-squared of 0.32, and adding the landscape factors improve this only very slightly, say to 0.34, management at the local scale will be more effective for conserving these species.

Thank you. We improved the analysis based on this suggestion and the new grouping of the categories. Our new approach involves the following changes:

- We only create single-predictor models for variables fitting the new definition of *landscape* (i.e. past and present landscape heterogeneity and cover of arable and open land).
- We fit new glms using landscape (past heterogeneity, past arable, present open), local (previous local model but now including grassland size and heterogeneity) and region (county), landscape+local, landscape+region, local+region. Adjusted R-squared values, along with chi-square tests comparing the full model with each group of variables removed allowed us to compare the relative and additional value of including each group of variables.

The new methods are described in **Lines 296-343** and results are added at **Lines 132-158, Table 3, Supplementary Tables 3-6**. It was true (unsurprisingly looking at the previous analysis) that landscape variables added a significant, but very small amount of added explanatory power to the model. We discuss this result at **Lines 198-203, 215-221**. We have also toned down the mention of the time lag response in the abstract (**Lines 28-39**).

(c) The authors do not specify if and how they tested for pairwise correlations between all the predictors in the 'full' model. I'm wondering if high collinearity between past landscape heterogeneity and one of the other predictors is the reason why the sign of the coefficient for this factor switches when it is included in the big model with all the other factors. The authors note this switch but do not offer an explanation for it. Might there also be interactions between 'region' and one or more of the other predictors? The authors didn't say if they considered potential interactions.

We did perform correlation tests between the variables before analysis, apologies for not documenting this in the original manuscript. Aside from the non-independency of the same variables across time period (i.e. historical arable and modern-day arable), past and present landscape forest cover was strongly correlated with other landscape variables and was therefore

removed prior to analysis. The highest correlation of variables used in the same model was 40% between grassland patch size and grassland heterogeneity (**Supplementary Table 9**).

We did not consider potential interactions in our model. It is likely that there are some interactions with local and landscape variables and region, as there are clear regional differences in landscape trajectories (**Figure 1**). However, interaction terms for each of the 14 counties with each of the local and landscape variables (or even a subset of them) would result in model outputs that would mean very little to most people, and be difficult to interpret even for those who are acquainted with the different regions of southern Sweden. Therefore, by including Region in the model, rather than for example including it as a random term in a mixed model, we highlight that there is a strong regional variation in grassland specialist richness. Together with the apparent regional differences in land-use change during the 20th century, we infer that these differences in combination with local and landscape variables have strong effects specialist richness. The new analysis supports this very nicely, so thank you again for the suggestion.

(2) Regarding use of the term ‘biodiversity’

“Biodiversity” is a ubiquitous term in ecological research but it is usually poorly defined.

Throughout the paper the authors make conclusions and interpretations about ‘biodiversity’ when what they are actually measuring is ‘species richness of grassland specialist plants’. This may seem trivial, but it is actually very important. If we really want to understand how landscape change is affecting plant and animal communities, we need to be specific about what we are measuring. In this case, we do not know how the changes on the landscape have related to ‘plant biodiversity’, because that would include ALL plant species, and would involve measuring not only taxonomic, but functional and genetic diversity as well. By choosing only grassland specialist species, the authors are making a value judgement up front that these are the particular species we should care about most. This should be made clear throughout the manuscript, preferably by avoiding the term biodiversity in favour of what was actually measured: richness of grassland specialist plants.

We agree. We have removed many cases of the term ‘biodiversity’, for example at all times when directly referring to our own data and results (entire manuscript). Where we retain the word, it is when introducing the global problem of land-use change effects on biodiversity (e.g. **Line 28, 53**), and when ‘scaling up’ our findings and incorporating them into the existing literature (e.g. **Line 228**).

Line by line comments:

Title: the title is not specific enough, and also misleading. Your data do not show that past habitat loss threatens plant biodiversity. Your data show that the species richness of grassland specialist plants is positively correlated with past landscape heterogeneity. I object to using the term ‘biodiversity’ when what the study has tested is species richness of semi-natural grassland specialist vascular plants. See overall comments above. The title should emphasize the true novelty of this study, which is assessing past and present landscape composition and heterogeneity at an unprecedented spatial scale, and testing for relationships between landscape composition/heterogeneity and plant community data from a massive set of on-the-ground surveys. Suggested titles:

Land-use change across a super-region and links to the diversity of grassland specialist plants
 Super-regional land-use change over several decades and its effects on the grassland specialist flora
 Quantifying land-use change across a super-region and testing its effects on the grassland flora

We liked the second suggestion, and have modified this to fit within the journal’s specifications. The new title is “Super-regional land-use change and effects on the grassland specialist flora” (**Line 1**). The ‘several decades’ part that we removed is mentioned in the abstract. Thanks!

Abstract:

Line 38: a decline in grassland specialist plant species. There might actually have been an INCREASE in total plant species richness (due to colonization by disturbance tolerant species, or weeds associated with crops or urban land use). It is important to be clear about this.

Line 39: again ‘biodiversity’ is misleading. “present-day richness of grassland specialists” is honest.

We agree, and there is of course plenty of evidence that total species richness can be stable or even increase over time. We have changed the wording here to reflect that we only consider specialist species (**Line 34-35**)

Main:

Line 72-76: Yes – this is well put and aptly describes the need for quantifying historical land-use/land-cover change at regional spatial extents.

Thank you.

Line 87: Why “landscape and spatial”? Why not just “landscape”? The term ‘spatial’ to me implies some kind of spatial autocorrelation analysis, which is not included here.

This has now been changed, see explanation above.

Line 91-94: again, ‘biodiversity’ is misleading. Suggest “diversity of grassland specialists”.

As explained above, we have gone through the manuscript to check our use of the term.

Line 96-98: it is a little strange to me that the ‘open’ category includes both grasslands AND urban land use – which seem quite starkly contrasted to me – but not arable land (which also, in many people’s understanding, is ‘open’). It might help to more explicitly define ‘open land’ where it is introduced, around line 79. Here, it would be easier for me to understand what is going on if you explained what replaced the open land. e.g. is it “... predominantly driven by a loss of semi-natural grassland cover due to conversion to arable land and afforestation” ?

We briefly define what open land is (mostly grasslands, but also urban and wetlands) in the methods and results summary at the end of the introduction (**Lines 86-89**). We give much more detail in the methods, alongside how land use is interpreted from the historical maps (**Lines 238-252**). In the now separate and extended results and discussion, we go further into detail about the different trajectories of land-use change and how they relate to the prevailing agricultural shifts that have taken place in Europe during the 20th century (**Lines 116-122, 173-182**).

Line 103: “This is due to a decline in arable land at the expense of more modern grassland and urban land uses”. How can you tell this from your data, which lumps modern grassland, semi-

natural grassland, and urban land use all together in the ‘open’ category?

As semi-natural grassland is by definition a historical land-use created by long term, low intensity management (**Lines 56-57**), we do not see that it is possible that this can make up the new areas of open land in landscapes where arable land has declined. We have previously shown that historical arable land that is no longer ploughed largely becomes low-biodiversity modern grassland types and to a lesser extent urban land use (Cousins et al. 2015, *Ambio*, doi: 10.1007/s13280-014-0585-9). The new opening paragraph of the discussion is more explicit in the changes that have taken place, citing both our previous study as well as those that describe broad changes in agricultural land use in Europe (**Lines 173-182**).

Figure 2: Is the beta reported correct? The number given in the text is different.

The figure in Figure 2 was correct and the figure in the text has now been changed (**Line 156**). Many thanks for noticing this.

Line 130: “by a very similar amount”. This is a bit vague. It looks to me like including all the variables explains a further 8.5% of the deviance explained by either local or landscape sets alone. I really think Supplementary Table 4 should be presented in the main text.

This has been changed due to the new analysis. The new version of the table is now presented in the main text (**Table 3**).

Line 134: parameter estimate here in text given as: 0.987. In Figure 2 it is given as: 0.991.

Changed. See above.

Line 138: “The loss of semi-natural grassland habitat has had a negative effect on biodiversity”. This is not accurate or precise. Your data show: OPEN habitats (including semi-natural grasslands, but also, wetlands, urban land use) have declined. Your data show: semi-natural grassland patches surrounded by more ‘open’ land use tend to have higher numbers of grassland specialist species

(although, the amount of variation explained is small in comparison to the effect of the AREA of the semi-natural grassland patch AND you don't actually have a measure of change in the number of grassland specialist species over time). Your statement should be: "The loss of open habitats may lead to a decline in the species richness of grassland specialist plant species in semi-natural grasslands."

We have now changed this sentence to: "This loss of semi-natural grassland habitat appears to have had a substantial negative effect on the grassland specialist flora" (**Lines 187-188**). In doing so, we have toned down the statement, although not to the extent suggested. In the (now) preceding paragraph, we explain why we think that the loss of open land was largely driven by losses of semi natural grassland, losses that were probably much larger than the total loss of open land. It is exactly because area of semi-natural grassland patch is by far the most important predictor of grassland specialist richness that we believe that the inferred loss of grasslands has led to declines in such species across the study region. This is explained in the following two sentences (**Lines 190-196**).

Line 144: present day richness of grassland specialists (NOT biodiversity). Also – yes I agree that the past landscape heterogeneity was a stronger predictor of grassland specialist species richness, but BOTH were quite weak compared to the effect of grassland area and grassland heterogeneity. Therefore, I expect it would be better (not to mention easier!) to manage for larger, more heterogeneous grassland patches, rather than trying to manage heterogeneity on a landscape scale!

We completely agree. It was not our intention to suggest that managed heterogeneity in the landscapes should be a priority to support biodiversity in grasslands. Rather, the (albeit small) effects of historical landscapes on present day specialist richness means that there could be time-lagged responses to land-use change and therefore that future losses might be expected. Conservation measures could help to reduce or stop these losses, but such measures should of course be focussed on grassland habitat itself. We have now rewritten this part for clarity (**Lines 198-203**).

Line 152: Yes, this change in sign of the correlation between landscape heterogeneity and grassland specialist species richness when in the full model compared to when alone is interesting. Is it because past landscape heterogeneity is correlated with one of the other predictors? (so the sign of the coefficient changes when the correlated predictor is included?). Or is it because the effect of

heterogeneity is different between regions, and perhaps an interaction term should have been included? I would like to know how much correlation there is between all the pairs of predictor variables.

See response to main comment along the same lines.

Line 156-163: I find this whole section very vague. HOW could this lead to “inaccurate or erroneous policy decisions and misdirected conservation management efforts”? Can you provide an example?

Line 159: “Local factors were shown to be most important predictors of grassland diversity, but we found that landscape and spatial data can provide useful local context and are more straightforward to collect at large spatial scales when site-by-site visits are not feasible”. Okay, but two of your “landscape” factors (in fact, the 2 with the strongest influence) are in my opinion actually “local” factors – the area of the grassland patch itself, and habitat heterogeneity within that patch. See my comment above re: presenting results in a way that allows us to measure the importance of the landscape-scale openness, heterogeneity, or arable cover once these local variables have been accounted for.

Line 161: “Such information could be a useful first step in consolidating national-scale policy with appropriate local-scale conservation practice.” Again, it is very unclear to me what you are suggesting here. Earlier in this paragraph you stated that places with different historical landscape conditions and landscapes in different REGIONS changed in different ways – which makes it dangerous to extrapolate patterns across different historical conditions or different regions. Now you are saying that national-scale policy should be consolidated. Do you mean that national-scale policy needs to be tailored such that different conservation measures are promoted in different regions and/or for landscapes with different historical conditions? On the other hand, in the previous paragraph you suggest that higher landscape heterogeneity ACROSS THE WHOLE REGION could promote richness of grassland specialists. So, should national-scale policy aim to increase landscape heterogeneity everywhere?

We have rewritten this paragraph to address these three points (**Lines 205-221**). We have toned down the (vague) message, as well as being more specific to how our results fit into the literature and could apply to conservation management. First we describe the well-known example of agri-environmental schemes that are designed at high governmental levels and applied at local levels, achieving mixed results often depending on landscape and regional context. We then add that based on our results, information describing the landscape and regional context can be of use in conservation management.

Line 169: “It is clearly imperative that remaining semi-natural grassland habitat is protected and appropriately managed to promote biodiversity given alarming rates of continued habitat conversion.” Okay, it is clearly imperative that IF we want to maintain richness of grassland specialists in semi-natural grasslands, THEN we should protect and manage these remaining grasslands. But I don’t think your scientific results lead to this conclusion – this is a value judgement.

Yes, but this statement is directly related to our study and its findings, and we think that conserving richness of species that are specialised to threatened and disappearing habitats is a shared value among many. We also think that the final paragraph of a manuscript is a relevant and appropriate place for such a statement, and have therefore not changed or removed it (**Lines 223-231**).

Tables:

Table 1: the asterisk should be beside “Present landscape open” instead of “Past landscape open”. Again, seems to me the area of the grassland patch, and the heterogeneity within this patch are actually ‘local’ predictors...

Thank you for spotting this error. The now local predictors grassland area and heterogeneity have been removed from the table (**Table 1**).

Table 2: change “species richness” in caption to “species richness of grassland specialists”. Also, the caption should note that region (county) is also included in the model.

Changed (**Table 2**).

Methods:

Line 325: was each grassland surveyed exhaustively, or did they just do surveys of PLOTS within the grasslands? e.g. this is total richness of grassland specialists in the whole grassland patch, or ‘species density’ of grassland specialists within a consistent same-sized survey plot? I think it is the former but it would be good to specify.

The surveys were of the whole grassland, although we cannot comment on how exhaustively each species was searched for. We have changed the sentence, which now begins “The whole of each inventoried grassland...” (**Line 260**)

Line 330: how could surveyors know what portion of the grassland had been fertilized or had seeds added? Did they interview land managers?

These values were based on visual estimations, as mentioned in the text (**Line 274**).

Line 331: “the fraction of openness in the grassland” – this is very confusing to me, especially when the term ‘open land’ is already being used on a regional scale to indicate grassland/wetland/urban land. By “fraction of openness” do you mean the fraction that is not invaded by woody trees or shrubs? It would be good to clarify this.

Yes, it is the fraction of land not covered by trees or shrubs. We have now clarified this (**Lines 262-268**).

Line 331: “30 categories of habitat”... so are these essentially finer-scale habitat definitions within the ‘open’ category? It might help to give a few examples (and reference Supplementary Table A1.

We have now updated this as suggested (**Lines 281-283**).

Line 335: by “indicator species”, do you mean, at least one of the 56 semi-natural grassland specialists?

Yes, thank you for noticing this. Now changed (**Line 286**).

Line 345: “plant biodiversity” – see overall comments. This study is not analyzing plant biodiversity. Could change to “diversity of grassland specialists”, or “species richness of semi-natural grassland specialists”

Now changed to Grassland specialist plant diversity (**Line 296**).

Line 352: shouldn't the number be 8, not 6, for the single-predictor models? Predictors are: (1) area of the surveyed grassland, (2) habitat heterogeneity of the surveyed grassland, (3) present landscape heterogeneity within 1000m buffer, (4) past landscape heterogeneity within 1000 m buffer, (5) present percentage open cover within 1000m buffer, (6) past percentage open cover within 1000m buffer, (7) present percentage arable cover within 1000m buffer, (8) past percentage arable cover within 1000m buffer

Yes, apologies for this oversight. Although now it is actually six models since we changed the variable groupings (**Line 305**).

Line 354: specialist species richness

Changed (**Line 292**).

Line 359: which type of r-squared did you use, and was it adjusted r-squared? Did you consider presenting the AIC for each model? What is the R² (or AIC) of a null model with intercept and region only?

We chose only to present R-squared. Together with the Chi-square comparison of the full vs. nested models we judge that this is sufficient to compare models and the contribution of the different sets of variables. We used adjusted R-squared across all analyses (**Lines 314, 339, Tables 2 & 3**).

Line 369: it is confusing here how ‘landscape’ versus ‘local and management’ were defined. If I understand correctly, the total area of the surveyed grassland, and the heterogeneity of different fine-scale habitat types within that patch, were considered ‘landscape’. But then the area of two

specific fine-scale habitat types within that patch ((1) semi-natural dry grassland and shrubland on calcareous substrates and (2) Fennoscandian species-rich dry-mesic lowland grassland) were considered ‘local and management’ predictors. Why would not grassland area and heterogeneity within the grassland be considered ‘local’?

See responses to earlier comments on the same subject

Also – wouldn’t there be a negative correlation between the area of the 2 fine-scale grassland types (e.g. Fennoscandian species-rich dry-mesic lowland grassland) and fraction improved grassland? If there is a greater area of improved grassland, wouldn’t there be less area of these specific semi-natural grassland types? Did you check for collinearity between the predictors before putting them all into the one large model?

We checked for collinearity prior to model building and the results are now documented in **Supplementary Table 9**, see response to earlier comment. In theory these three variables could be negatively correlated as suggested, but because of the fact that not (or very few) grasslands had two or more of these ‘habitat types’, many had none at all and because of the sheer number of grasslands in the database, the correlation was very small.

Line 382: “Chi-squared tests were then performed to evaluate the additional explanatory power contained within the third, full model compared to the first landscape and spatial and the second local and management models (R function: anova).” I don’t see these results reported anywhere... oh, now I see it’s in Supplementary Table 4. I think this table should be in the main text.

Now in the main text, as suggested earlier (**Table 3**)

Supplementary Information:

Line 19: insert “and” before “total grassland area”

Table A1: Alkaline fens

Supplementary Table 4: I really think this should be presented in the main text.

Supplementary Table 6:

Epipactis palustris

Hypochaeris maculata

Many thanks for checking the supplementary material so carefully, these errors have been corrected.

REVIEWERS' COMMENTS:

Reviewer #1 (Remarks to the Author):

Review

Auffret et al

Super-regional land-use change and effects on the grassland specialist flora

Thank you for revised version, I am happy with the responses given to my previous review (I was Reviewer 1). I am still convinced that it is an important contribution, providing information about land-use changes in an unprecedented scale. I have few remaining comments.

Introduction – it is still a bit vague from the introduction why category „open land“ is a good quantifier of biodiversity-relevant landscape changes, as the link between grasslands and open land is not clearly stated. In only one sentence it can be brought out more clearly that the amount of open land is related to cover of semi-natural grasslands and loss of open land expresses loss of semi-natural grasslands.

line 66-67 – please rephrase so that the sentence would not hint that time-lags occur due to more heterogeneous landscapes (as it currently reads).

line 93 and 99 – please also shortly state which were the regional and local variables (as done for landscape variables).

line 100 – sentence starting „Local factors are...“ is vague, as it is not instantly clear whether you are already talking about your own findings or general expectations. Add “We found” or something similar at the beginning.

line 132 – Drivers of grassland specialist richness – clarify that you are focusing on plants

line 200 – „best spent“ instead of what? Current wording hints the conflict between preventing further declines and improving local factors. I understand that the authors argue between improving landscape conditions vs improving local conditions, but it can be phrased more clearly.

lines 215-221 – Without explaining what is considered under landscape vs. local factors, this section can give misleading information. In current study, „local factors“ also involve habitat area. While it is OK to classify it as a local factor, area of habitat has often been considered as a landscape factor in other studies, whereas „local factors“ in other studies mostly include site-specific environmental conditions. Thus, instead of just discussing undefined local vs. landscape factors, I'd suggest more detailed recommendations for conservation actions e.g. stating that increase of area and quality of local habitats is necessary.

Supplementary T8 – check “Pulsatilla pulsatilla” – is Pulsatilla pratensis meant here?

Supplementary Figure 1 – what is marked with yellow on panels c-e? If it is respective habitat cover, then forest (panel e) does not seem to be covered with yellow?

Reviewer #2 (Remarks to the Author):

Review of “Super-regional land-use change and effects on the grassland specialist flora” (revised manuscript) by Auffret et al. for Nature Communications

July 1, 2018.

I find the manuscript very much improved and my main suggestions and concerns have been addressed.

1. In response to my comment, and the same comment from the other reviewer, the authors have reclassified the predictor variables into groupings based on scale (landscape or local).

2. The revised analysis now clearly compares models including different sets of predictors and presents the adjusted R squared values for each one, making it easy for the reader to gauge how much additional variation the landscape predictors are adding once local variables and the region in which the grassland is found have been accounted for.

3. The authors now include in the supplementary material the results of pairwise correlation tests between the non-categorical predictor variables, and describe which variables were not included due to strong correlations.

4. The manuscript now is specific about what portion of grassland plant diversity is actually being measured – namely, the species richness of grassland specialists.

I also found the methods to be much clearer, and the interpretation of the results presented in the discussion and implications for conservation management better explained than before.

Small, line-by-line comments:

Line 310: Past and present forest cover were NOT used

Line 329: insert "grassland specialist" before "species richness"

Reference number 14 has an error.

The references in the Supplementary Information are cut off – perhaps just an artefact of the conversion to pdf?

Response to referees.

Thanks again to both referees for the excellent comments during the review process. We now respond to the second round of review. Our responses are in red, and directions to changes in the manuscript refer to the tracked-changes version.

Alistair Auffret and co-authors.

REVIEWERS' COMMENTS:**Reviewer #1 (Remarks to the Author):**

Review

Auffret et al

Super-regional land-use change and effects on the grassland specialist flora

Thank you for revised version, I am happy with the responses given to my previous review (I was Reviewer 1). I am still convinced that it is an important contribution, providing information about land-use changes in an unprecedented scale. I have few remaining comments.

Thank you.

Introduction – it is still a bit vague from the introduction why category „open land“ is a good quantifier of biodiversity-relevant landscape changes, as the link between grasslands and open land is not clearly stated. In only one sentence it can be brought out more clearly that the amount of open land is related to cover of semi-natural grasslands and loss of open land expresses loss of semi-natural grasslands.

We could not find an appropriate place for such a sentence. The final paragraph of the introduction, containing as it must a brief account of all our methods and findings means that such a sentence would disrupt the flow of the text. What we have done is to be more specific when briefly mentioning that open land is largely interpreted as semi-natural grassland (Lines 88-89), while also referring the reader to the Methods where this information is explained in more detail and in the

context of the description of the historical maps (Lines 243-257). Further down in this paragraph we already remind the reader that the loss of open land is interpreted as a loss in semi-natural grassland when relating land-use change to present day specialist richness (Lines 105-109).

line 66-67 – please rephrase so that the sentence would not hint that time-lags occur due to more heterogeneous landscapes (as it currently reads).

We have now rephrased the sentence to speak in more general terms of the landscape influence on time lagged biodiversity responses, rather than specifically heterogeneity (Lines 66-68). *“As a result of processes occurring at landscape scales, communities can experience time lags of decades, or even centuries, before predicted future diversity losses.”*

line 93 and 99 – please also shortly state which were the regional and local variables (as done for landscape variables).

Again, we do not think there is room for this information in this paragraph, which should only contain “brief summary of both the results and the conclusions”. The landscape variables are only described briefly “landscape variables *including* past and present land use and heterogeneity surrounding each grassland” (Lines 97-98, italics added here only), and we think that the current description of local variables “local variables describing the character of the focal grassland and its management” (Lines 99-100) is sufficient here. Full information is of course found in the Methods and the Results.

line 100 – sentence starting „Local factors are...“ is vague, as it is not instantly clear whether you are already talking about your own findings or general expectations. Add “We found” or something similar at the beginning.

“We find...” added (Line 102)

line 132 – Drivers of grassland specialist richness – clarify that you are focusing on plants

Changed to “Drivers of grassland specialist plant richness” (Line 134)

line 200 – „best spent“ instead of what? Current wording hints the conflict between preventing further declines and improving local factors. I understand that the authors argue between improving landscape conditions vs improving local conditions, but it can be phrased more clearly.

We are now clearer that our results suggest that the focus of conservation actions should be on local grassland management rather than the landscape.

Now changed to “...much stronger local effects on specialist plant richness suggest that resources would be best spent on maintaining, improving and restoring existing and abandoned pastures and meadows (which would in many cases also increases landscape heterogeneity), rather than focussing on improving the heterogeneity of the landscape *per se*.” (Lines 202-204).

lines 215-221 – Without explaining what is considered under landscape vs. local factors, this section can give misleading information. In current study, „local factors“ also involve habitat area. While it is OK to classify it as a local factor, area of habitat has often been considered as a landscape factor in other studies, whereas „local factors“ in other studies mostly include site-specific environmental conditions. Thus, instead of just discussing undefined local vs. landscape factors, I’d suggest more detailed recommendations for conservation actions e.g. stating that increase of area and quality of local habitats is necessary.

We have now improved this section. Although we have not stated that the increase in area and quality of local habitats is necessary (as this is already discussed in the preceding paragraph), we make it clear that it is not only landscape-scale data that can be readily available, and useful for gauging the biodiversity value of habitats where site visits are not possible (Lines 219-224).

Supplementary T8 – check “*Pulsatilla pulsatilla*” – is *Pulsatilla pratensis* meant here?

Changed to the correct name *Pulsatilla vulgaris* (Pasqueflower).

Supplementary Figure 1 – what is marked with yellow on panels c-e? If it is respective habitat cover, then forest (panel e) does not seem to be covered with yellow?

The yellow shading on the maps (panels c-e) is arable land (see lines ... in main document). There is indeed some yellow in panel e. More information has been added to the figure text.

Reviewer #2 (Remarks to the Author):

Review of “Super-regional land-use change and effects on the grassland specialist flora” (revised manuscript) by Auffret et al. for Nature Communications

July 1, 2018.

I find the manuscript very much improved and my main suggestions and concerns have been addressed.

1. In response to my comment, and the same comment from the other reviewer, the authors have re-classified the predictor variables into groupings based on scale (landscape or local).
2. The revised analysis now clearly compares models including different sets of predictors and presents the adjusted R squared values for each one, making it easy for the reader to gauge how much additional variation the landscape predictors are adding once local variables and the region in which the grassland is found have been accounted for.
3. The authors now include in the supplementary material the results of pairwise correlation tests between the non-categorical predictor variables, and describe which variables were not included due to strong correlations.
4. The manuscript now is specific about what portion of grassland plant diversity is actually being measured – namely, the species richness of grassland specialists.

I also found the methods to be much clearer, and the interpretation of the results presented in the

discussion and implications for conservation management better explained than before.

Many thanks. We are glad that you approve of the changes made, and agree that they have resulted in a better manuscript

Small, line-by-line comments:

Line 310: Past and present forest cover were NOT used

“not” added (Line 315). Thank you.

Line 329: insert “grassland specialist” before “species richness”

Added (Line 335).

Reference number 14 has an error.

Fixed (Line 403).

The references in the Supplementary Information are cut off – perhaps just an artefact of the conversion to pdf?

Yes, this was due to the manuscript submission system and is not a problem with the document itself.